# Early Trauma Leaves No Social Signature in Sanctuary-Housed Chimpanzees (*Pan troglodytes*)

**DOI:** 10.3390/ani13010049

**Published:** 2022-12-22

**Authors:** Edwin J. C. van Leeuwen, Bernadette M. C. Bruinstroop, Daniel B. M. Haun

**Affiliations:** 1Animal Behaviour and Cognition, Department of Biology, Utrecht University, Padualaan 8, 3584 CA Utrecht, The Netherlands; 2Department of Comparative Cultural Psychology, Max Planck Institute for Evolutionary Anthropology, Deutscher Platz 6, 04103 Leipzig, Germany

**Keywords:** chimpanzees, welfare, social deprivation, trauma, coping, sanctuary

## Abstract

**Simple Summary:**

Negative experiences in early life, such as the loss of the mother, can have negative and long-lasting consequences on social functioning in adolescence and adulthood in both humans and other socially living animals, such as our close relative the chimpanzee. Recent studies indicate that zoo-housed chimpanzees may be socially scarred for life after such early trauma in terms of whom they like to be near (social proximity) and whom they entrust keeping their body clean (grooming). In light of these findings, the current study investigated whether the same effect would be identified among orphaned chimpanzees living in social groups in large, forested enclosures in an African sanctuary. Overall, the orphaned chimpanzees were found to be socially indistinguishable from their counterparts who did not lose their mother and were born and mother-reared in the sanctuary. These results suggest that sanctuaries can be valuable rehabilitation centres for orphaned chimpanzees, facilitating chimpanzees’ potential to cope with early life adversities.

**Abstract:**

Negative early experiences can have detrimental effects on social functioning in later life, both in humans as well as in other socially-living animals. In zoo-housed chimpanzees, recent evidence suggests that there may be a lingering signature of early trauma on individuals’ social interaction tendencies as measured by social proximity and grooming. Here, we address whether a similar effect would be observable in chimpanzees living under semi-wild conditions in an African sanctuary. By analysing party size, close proximity and social grooming, we show that in this specific sanctuary, chimpanzees that suffered early trauma (*n* = 42) were socially indistinguishable from chimpanzees who were born and raised by their mothers in the sanctuary (*n* = 36). Our findings indicate that chimpanzees may not be irreversibly affected by early social trauma, possibly owing to rehabilitation in stable social groups in a semi-natural environment. Beyond identifying sanctuaries as valuable rehabilitation centres for orphaned chimpanzees, this study demonstrates a remarkable social flexibility in one of our closest living relatives.

## 1. Introduction

“Chimpanzees have suffered so much pain and trauma at the hands of humans... yet they still have the grace to forgive us.”—Sheila Siddle (founder of the Chimfunshi Wildlife Orphanage).

Early life adversities can have long-term effects on social competence in humans [1,2,3], but also in non-human primates. Plausibly owing to a combination of the impact of traumatic experiences on brain functioning (likely mediated by elevated stress levels [4]) and a sub-optimal environment for social ontogeny [5,6,7], primates deprived of a nurturing upbringing may be cognitively and socially scarred for life [8,9,10,11,12,13,14]. Yet, primates possess a remarkable level of cognitive and social flexibility in the face and aftermath of detrimental circumstances. Both in humans [2,3,15] and chimpanzees [16] resilience to hardship has been observed in form of re-established social competence. For instance, after institutionalization, many orphaned children showed substantial recovery on socio-cognitive and linguistic tests once they were transferred into foster care [15], and chimpanzees with disrupted developmental trajectories in early life (i.e., pet-trade and entertainment business) have been found to respond as adequately to social alterations as their non-deprived counterparts [8]. Determinants of recovery comprise genetic and idiosyncratic factors, but also the post-institutional social environment including its potential to nurture individuals’ socio-cognitive capital [3,17].

To better understand when adversities develop into long-lasting versus recoverable social consequences, it is essential to build a corpus of naturally-occurring cases in which animals experience social trauma and document their behaviour in the aftermath thereof. In light of the unfortunate reality that chimpanzees are regular victims of human-induced trauma (e.g., pet-trade, bush-meat trade) [18], chimpanzees have been studied frequently, both as models for human social resilience and their well-being in its own right. To date, on the one hand, studies have demonstrated that the experience of traumatic events by chimpanzees can lead to the development of abnormal behaviour in which the chimpanzees show signs of posttraumatic stress disorder (PTSD), depression, anxiety disorder, obsessive-compulsive disorder and are less socially skilled [6,9,19,20,21]. These effects show great similarities with the effects of trauma on humans, in which traumatized children develop PTSD, depression and generalized anxiety disorder, and experience difficulty regulating, identifying, and expressing emotions [17,22]. Bradshaw et al. (2008) analysed several case studies of chimpanzees who sustained prolonged biomedical experimentation in deprived settings but were later housed in sanctuaries [23]. The analysis concluded that a diagnosis of Complex PTSD in chimpanzees is consistent with descriptions of trauma-induced symptoms as described by the DSM-IV and human trauma research. As a result, the authors illustrate how human psychological models of diagnosis and treatment might be approached in great apes.

According to a case study performed on chimpanzees in a sanctuary, chimpanzee behaviour resulting from deprivation of maternal care showed remarkable similarity with the behaviour of human infants and young children who did not receive adequate maternal care, including preoccupation with the body and body products, as well as stereotypic behaviours providing needed self-stimulation and/or self-soothing [24]. Additionally, studies that looked into the interaction of chimpanzees with conspecifics and/or humans and into their allogrooming activity have shown that chimpanzees deprived of a nurturing upbringing may be cognitively and socially scarred for life [8,9,10]. On the other hand, there are indications that chimpanzees can (partially) recover from suffered trauma. For instance, a case study showed that access to an enriched social environment might help chimpanzees recover from the loss of their mother [19]. Additionally, traumatized chimpanzees have been shown to benefit from therapeutic resocialisation [25] in combination with rehabilitation programs for social and natural behaviours [26].

A closer look at the impact of psychological trauma in humans reveals that trauma can have a wide range of recovery outcomes. Some humans become permanently disabled due to the experienced trauma, others may fully recover, and yet others may even achieve an enhanced, more adaptive level of functioning as the result of working through their traumatic experiences [24]. This latter aspect emphasizes the potential role of the post-traumatic (social) environment for the recovery process (also see [3]). In this light, it is relevant to note that studies reporting (long-lasting) negative effects of early life adversities on chimpanzees’ social wellbeing have typically been conducted on chimpanzees living in zoo-like settings or small atypical social groups in non-home-range countries (e.g., [8,9,10,27,28]). For instance, among zoo-housed chimpanzees in the USA, abnormal behaviours were reported to be more frequent in non-mother-reared individuals compared to their mother reared counterparts [28]. Furthermore, chimpanzees with relatively little exposure to conspecifics in the first 4 years of their lives were found to exhibit lower frequencies of grooming and sexual behaviours later in life: two essential social activities within chimpanzee communities [10]. Similarly, chimpanzees who were orphaned within the first two years of their lives experienced problems with socialising as adults—in particular, they held back from social grooming, an interaction which is vital for establishing and maintaining social relationships ([8] also see [29]). Lastly, it was observed that ex-laboratory and early long-term maternally deprived zoo chimpanzees were more restricted in their grooming associations compared to non-deprived zoo chimpanzees, leading the authors to conclude that “early maternal loss has lifelong effects on the social integration of chimpanzees” [9].

Interestingly, however, a longitudinal study on the short- and long-term effects of maternal loss on wild chimpanzees showed that immature chimpanzees who lost their mother at a young age may initially become highly stressed, but after two years after the loss of their mother, the orphans were found to be no more stressed than chimpanzees whose mother did not die [30]. The respective authors suggest that this may be because other chimpanzees care for or even adopt the infants (see [31]). This could mean that in order to help infant chimpanzees recover from the trauma of losing their mother, the presence of other (already familiar) chimpanzees might be important, possibly because these chimpanzees can take over the care for the infant and help them become physically and emotionally healthy individuals. Moreover, these findings suggest that a natural environment may aid in the recovery of chimpanzees who experienced (severe) adversities.

Here, based on the hypothesis that socially-deprived chimpanzees may be facilitated in their social recovery by a nurturing environment, we set out to explore the signatures of early life trauma on chimpanzees’ social lives in four large groups of semi-wild chimpanzees at the Chimfunshi Wildlife Orphanage Trust in Zambia. At Chimfunshi, chimpanzees live in large (47 to 190 acres) forested enclosures of which its ecology was assessed as highly suitable for chimpanzees (against the benchmark of the ecologies and needs of wild chimpanzees) [32]. Moreover, from the early founding days in the 1980′s, Chimfunshi embraced a policy in which rescued chimpanzees were allowed to breed following the rationale that social rehabilitation would be expedited by the chimpanzees starting and living amongst family units. As such, the respective groups of Chimfunshi chimpanzees comprise chimpanzees from all age classes (infants, juveniles, adolescents, adults, and elderly) and from a range of different families that even engage in the fission-fusion dynamics typical of wild chimpanzees [33,34] (for more information on the Chimfunshi sanctuary, e.g., see [6]).

When housing chimpanzees in captivity (or semi-wild conditions) it is important to consider the effects of the limiting space. A study by Duncan et al., (2013) showed that when chimpanzees are under high spatial density conditions in an outdoor environment, they use a tension-reduction tactic to limit their aggression [35]. This means the chimpanzees increase affiliative behaviours and decrease aggressive behaviours as a coping strategy [36]. In indoor environments, orphaned chimpanzees adopted a tension-reduction tactic to limit aggression, while chimpanzees living among family adopted a conflict-avoidance tactic. In the conflict-avoidance tactic, chimpanzees decrease all social interactions (both affiliative and aggressive) as a coping strategy [36]. Both groups showed an increase in abnormal behaviour under indoor high-density conditions. This suggests that chimpanzees might also use abnormal behaviour as an outlet for the stress resulting from spatial restriction [35]. Another study found that short-term increases in spatial density results in the conflict-avoidance strategy in female chimpanzees, but in the tension-reduction strategy among males. During long-term high-density circumstances, females increased their affiliative behaviour while aggression remained at a steady low level, partially supporting the tension-reduction strategy. Males showed both an increase in affiliative behaviour and a decrease in aggressive behaviour, which fully supports the tension-reduction strategy [36]. Both studies highlight the importance of taking into account the spatial density conditions of the chimpanzees in captivity. At Chimfunshi, the chimpanzees are housed under low-density conditions, which might facilitate their social recovery.

To study the possible effect of detrimental circumstances on the social lives of the Chimfunshi chimpanzees, we focused on three core dimensions of chimpanzee sociality—party size [37,38], close association (i.e., proximity), and grooming [8,38,39], including individuals’ respective signatures in their social networks. Specifically, we investigated whether chimpanzees who had suffered social trauma during early ontogeny (which in all cases led to the loss of their mothers) could be quantifiably identified in these behavioural dimensions. We conjectured that the semi-wild conditions of the Chimfunshi Wildlife Orphanage Trust may function as a buffer against the lingering impact of early trauma and facilitate chimpanzees’ rehabilitation process to the extent that they become socially indistinguishable from their typically-developed counterparts in the sanctuary.

## 2. Materials and Methods

### 2.1. Study System and Subjects

Data were continuously collected from March 2011 to March 2013 at the Chimfunshi Wildlife Orphanage Trust, a chimpanzee sanctuary in Zambia. Subjects comprised 78 chimpanzees across four groups (Group 1–4) in Chimfunshi’s “Project area” living in forested enclosures ranging in size from 47 to 190 acres (Figure 1). Chimpanzees at Chimfunshi stay outside overnight and only come indoors for supplemental feeding between 11.30–13.30. Approximately half of the chimpanzees were wild born (*n* = 42) and integrated into peer groups at the sanctuary. The other chimpanzees were mother reared within the social groups at the sanctuary (*n* = 36). For demographic details on the chimpanzees, see Appendix A.

Upon arriving at the sanctuary (average age 3.2 years), the rescued chimpanzees were typically placed under human care for a short period of time, after which they were integrated into an existing social group following local and circumstantial protocols. Unfortunately, no or unverifiable data are available on the exact conditions under which the individual chimpanzees arrived at the sanctuary, yet in all cases they came without their mothers. Once a group was evaluated to be stable (i.e., exhibiting typical chimpanzee behaviours without excess levels of aggression), new rescues were sorted into a new group. Subsequently, the stable groups were then moved from the Orphanage to the Project area, where the chimpanzees gained access to more spacious forested habitats (Figure 1) and started living under minimal-human-interference protocols, i.e., apart from one or two supplemental feedings, the chimpanzees were not disturbed by humans. Groups 1–4 were formed between 1984–1989, 1990–1994, 1995–1999, and 2000–2002, respectively.

### 2.2. Data Collection and Operational Measures

Data collection comprised focal following using a standardized protocol [37]. Subjects were quasi-randomly selected as focal subject by a trained observer (E) starting at one of 4 (one of 7 in the two larger groups) pre-assigned locations surrounding the enclosure and selecting the subject closest to the start location. Subjects were video-recorded (centred with a 2 m radius) continuously for 10 min. If the focal moved out of sight, data were only included when the total time the focal was in view exceeded 5 min. At the end of each focal follow, one scan sample was obtained by E panning from left to right. All chimpanzees observed during the focal follow and scan sample were counted to belong to the focal’s party composition. The next focal chosen was the closest chimpanzee to the previously recorded focal. Observations were carried out for one hour every day, alternatingly between 8:30–11:00 and 14:00–16:30. Only one video per subject per week was randomly selected to increase data independency, resulting in a total of 3002 focal follow videos for analysis (group 1–4, *n* = 765, *n* = 911, *n* = 635, *n* = 691, respectively).

From the videos, we derived party size, proximity and grooming using a standard chimpanzee ethogram (adapted from [40]). Party size was defined as the sum of individuals within a focal’s party composition (including the focal). Proximity was defined as being in a 1-m radius of the focal individual; direct passings within a 1 m radius (without a moment of paused locomotion), grooming or agonistic encounters were excluded from this category. Grooming was defined sensu Nishida et al. [40] and counted both when the focal provided or received grooming (i.e., directionality not considered here). Per day, a 1/0 sampling method was used (for each behaviour coded) to maximize data independency [41]. Prior to coding the videos, all members of the coding team demonstrated high inter-observer reliability with a lead coder (Cohen’s kappa ≥0.85). Videos were coded in INTERACT version 15.0 (Mangold International GmbH, Arnstorf, Germany) and Excel. Party size, proximity and grooming were measures with sufficient data for statistical analysis (*n* = 3002, *n* = 6064, *n* = 946, respectively).

For the analyses on social integration (“are wild-born chimpanzees more or less integrated in their respective social groups than sanctuary-born chimpanzees?”), we calculated social network indices with SOCPROG [42]. Here, and throughout the manuscript, we work under the assumption that chimpanzees who were born in the wild have suffered social trauma before arriving at the sanctuary in the form of at least being separated from their mothers). First, we extracted twice-weight association indices [41], both for the proximity and grooming data. The twice-weight index was chosen as it is the least biased when there is an increased possibility of observing individuals who were associated over those alone ([43] also see [44]). The twice-weight association index (AI) is calculated as:

*x*/(*x* + 2*y*_AB_ + *y*_A_ + *y*_B_)
where x = the number of sampling periods (days) in which individual A and individual B were associated, *y*_A_ = the number of sampling periods in which only A was identified, and *y*_B_ = the number of sampling periods in which only B was identified, and *y*_AB_ = the number of sampling periods in which both A and B were identified but not associated with each other. “Identified” refers to an individual being captured on video that day, either as a focal subject or as present in the subgroup of another focal subject. Second, the following social network attributes (SNas) per individual were extracted, both for the proximity and grooming data: Strength, Eigenvector-centrality, Reach, Clustering, and Affinity (see Table 1). These measures were chosen for they represent meaningful dimensions of individuals’ social embeddeness. “Strength” represents the summed (weighted) associations, “Eigenvector-centrality” the connectedness of an individual by co-measuring the connectedness of its direct associates, “Reach” the length of the shortest path to the least associated individual, “Clustering” the proportion of associates that are associated among each other, and “Affinity” the extent to which one’s neighbours associate themselves [41,45].

### 2.3. Data Analysis

First, we analysed whether party size was contingent on the focal’s origin (i.e., wild-born versus sanctuary-born) using Generalized Linear Mixed Models with Poisson error distribution and log link function (lme4 package: [46]). The full model consisted of the fixed effects origin (wild/sanctuary born), rank (*z*-transformed), age (*z*-transformed) and sex. Focal follow duration was included as offset term to control for observation effort. Furthermore, we included population size and number of family units with population as offset terms to control for demographically relevant factors possibly influencing party size. Lastly, we included the random intercepts for focal, day and population-identity, and the random slopes for rank and age nested in day. The null model resembled the full model, except for the omission of the fixed effect for “origin”. The effect of “origin” was tested by comparing the full to the null model with a Likelihood Ratio Test (henceforth LRT: [47]). Furthermore, an auxiliary analysis on a subset of the data (i.e., only the wild-born chimpanzees) was conducted to test if chimpanzees who were early—(<3 yrs) versus later—(≥3 yrs) orphaned—as proxied by their age on arrival at Chimfunshi—differed in their party sizes. Here, the hypothesis was that early-orphaned chimpanzees would be less social than later-orphaned [27,48], resulting in smaller average party sizes.

Second, the association indices (for proximity and grooming, respectively) were analyzed using hurdle models (to accommodate the numerous zeros reflecting absence of association). The hurdle models consisted of a Binomial part (logit link function) to model the likelihood of presence/absence (i.e., probability) of association, and a Gamma part (log link function) to model the non-zero dimension (i.e., magnitude) of associations. Both model types consisted of the fixed effects dyad.origin (wild-wild, wild-sanctuary, sanctuary-sanctuary), dyad.sex (female-female, male-female, or male-male), and dyad.age (subadult-subadult, subadult-adult, or adult-adult). To account for kinship effects, we further included a variable denoting whether or not the dyad was between family members (same.matriline yes/no) as fixed effect. Furthermore, we included population size as offset term. We included the random intercepts of population-identity, focal and partner, including all possible random slopes within focal and partner [49,50]. For the “grooming” analyses, we only included chimpanzees older than 6 years of age for reasons of biological relevance (i.e., younger chimpanzees do not frequently engage in grooming activities [7]). The full models were compared with reduced models (using LRT [47]) to assess the effect of “dyad.origin”. Moreover, to gauge the reliability of the results, we assessed the stability of the applied models by excluding subjects one at a time and comparing the model estimates derived for these data with those derived for the full data set (indicating no influential subjects to exist).

Lastly, for the individually-derived social network attributes, we permuted (*n* = 1000) “origin” across individuals (keeping the ratio constant) to test whether the obtained distributions of network indices across individuals with different origin (sanctuary-born vs. wild-born) were significantly different from random distributions across individuals. Here, again, we additionally tested for the possible differential effects on early—(<3 yrs) versus later—(≥3 yrs) orphaned chimpanzees following the hypothesis that early-orphaned chimpanzees would be less embedded in their social networks [27,48].

All models were fitted in R (version 3.3.3: [51]) using the function “glmer” of the R package lme4 (version 1.1-12: [46]). We considered *p*-values less than 0.05 as significant and corrected for multiple testing using Bonferroni–Holm corrections [52].

## 3. Results

### 3.1. Party Size

Party size was not different for individuals of either origin (LRT: χ^2^ = 2.44, *p* = 0.118; Estimate ± SEM = −0.105 ± 0.066). Wild-born individuals congregated in parties of $\tilde{x}$ ± SD = 6.46 ± 4.37 group members, whereas sanctuary-born individuals formed parties of 7.36 ± 4.89 group members (Figure 2). The effect of origin was not obviously different across the four populations (interaction population & origin: LRT: χ^2^ = 1.013, *p* = 0.798). The early- and later-orphaned chimpanzees did not differ with respect to party size (LRT: χ^2^ = 0.20, *p* = 0.651; Estimate ± SEM = 0.028 ± 0.062). Early-orphaned chimpanzees were observed on average (±SD) in parties of 6.19 ± 4.19 chimpanzees, whereas later-orphaned chimpanzees were observed on average (±SD) in parties of 6.63 ± 4.53 chimpanzees.

### 3.2. Proximity

Regarding the existence of proximity associations (yes/no), wild-born individuals did not have fewer connections with group members than sanctuary-born individuals (LRT: χ^2^ = 0.698, df = 2, *p* = 0.701). On the contrary, dyads consisting of (only) wild-born individuals were slightly more often connected than the other dyad-types (wild-wild: 0.86%; wild-sanctuary: 0.68%; sanctuary-sanctuary: 0.63% of dyads).

Similarly, the extent to which the associated dyads engaged in close proximity was not significantly dissimilar across the different dyad-types (LRT: χ^2^ = 2.26, *p* = 0.323; $\tilde{x}$ ± SD wild-wild: 0.037 ± 0.029; wild-sanctuary: 0.040 ± 0.055; sanctuary-sanctuary: 0.032 ± 0.041). Furthermore, the individual-level social network metrics showed no obvious differences between wild- and sanctuary-born individuals based on proximity (Table 1 & Figure 3a). Similarly, the network metrics showed no obvious differences between early- and later-orphaned chimpanzees (all *p* > 0.23).

### 3.3. Grooming

Regarding the existence of grooming associations (yes/no), wild-born individuals did not have fewer relationships with group members than sanctuary-born individuals (χ^2^ = 1.150, df = 2, *p* = 0.563). Dyads consisting of (only) wild-born individuals were more often connected than the other dyad-types (wild-wild: 0.41%; wild-sanctuary: 0.19%; sanctuary-sanctuary: 0.14% of dyads).

Similarly, the extent to which the associated dyads engaged in grooming was not significantly dissimilar across the different dyad-types (χ^2^ = 0.003, df = 2, *p* = 0.999; $\tilde{x}$ ± SD wild-wild: 0.024 ± 0.022; wild-sanctuary: 0.035 ± 0.038; sanctuary-sanctuary: 0.033 ± 0.044). Furthermore, similar to the proximity results, the individual-level social network metrics showed no obvious differences between wild- and sanctuary-born individuals based on grooming (Table 1 & Figure 3b). With respect to the differences in network metrics between the early- and later-orphaned chimpanzees, again, no obvious effects were detected (all *p* > 0.15), except for a difference in the “reach” of grooming associations (*p* = 0.028; corrected for multiple testing [52]), with the later-orphaned chimpanzees having a slightly larger reach ($\tilde{x}$ ± SD = 0.048 ± 0.042) than the early-orphaned chimpanzees (0.045 ± 0.027).

## 4. Discussion

In this study we investigated whether chimpanzees with an early social trauma due to the loss of their mother suffered social consequences later in their life. We did this by testing whether the orphaned chimpanzees—now adolescents and adults—were socially (in)distinguishable from their group members who have experienced a normal upbringing with their mothers. We tested this in a sanctuary setting in which the orphaned chimpanzees were habituated into existing groups of chimpanzees upon arrival or shortly thereafter. When these groups were deemed stable by the local caretakers, they were moved to a remote area of primary miombo forest with sufficient space to engage in fission-fusion patterns, to find fresh leaves and trees to spend the nights in, and to forage on food resources typical of wild chimpanzee diets [32].

By analysing the average party sizes that the chimpanzees roamed in and their tendencies to be in close proximity to others and groom their group members, we consistently found an absence of differences between the orphaned chimpanzees and their typically-developed counterparts. Similarly, the social network metrics, which reflect the chimpanzees’ social embeddedness in their groups, did not show any obvious differences between the two groups either. If the early social trauma suffered by the orphaned chimpanzees would have had lingering effects, as has been observed in other groups of chimpanzees (e.g., [9]), we may have observed a signature in their social behaviour one way or another, be it in their tendencies to isolate themselves from the group (i.e., smaller party sizes) or in their refraining from seeking bodily closeness or contact with group members (i.e., close proximity and grooming) [8,29]. In other words, the strength of these findings lies in the consistency of an absence of differences, especially given our fairly large sample size.

Nonetheless, this study is not conclusive, as it is impossible to pinpoint with certainty which factors have caused the chimpanzees’ social recovery. In fact, we do not even have data on the extent to which the respective chimpanzees suffered by the loss of their mother—instead, we work under the assumption that chimpanzees suffer when becoming orphaned and that a certain rehabilitation process is needed to re-socialize them. Support for this assumption is relatively abundant, both from ethnographic descriptions of chimpanzee life in the wild [33] as well as detailed studies of the effect of trauma on social behaviour such as grooming, abnormal behaviour such as coprophagy, sexual behaviour and activity levels of chimpanzees in zoo-settings [8,9,10,28,29]. In zoo-settings, typically, the signatures of early trauma on chimpanzees’ social behaviour remain detectable in later life, even into adulthood. These signatures include a smaller grooming network, a wider repertoire and elevated levels of abnormal behaviour (but reduced coprophagy) and a reduced normal activity level. Early trauma in chimpanzees can thus have severe implications for their welfare as well as for colony and population management in general [8,9,19,27]. This link has particularly been established for chimpanzees who lose their mothers at a relatively young age, so-called early deprivation ([27,48], idem for humans, see [1,2,3]). The reason for this effect might be that the youngsters have missed the opportunity to establish secure attachments to their mothers, from which they typically derive a certain extent of self-esteem and self-efficacy [48].

Based on our results, we tentatively conjecture that the semi-wild conditions under which the Chimfunshi chimpanzees are being rehabilitated, both in terms of housing (e.g., large, forested enclosures, foraging options and outside sleeping) and socio-demographically (i.e., groups with species-typical group compositions) may contribute to the success of their social recovery. In addition to our overall findings, we did not detect obvious differences in party size measures and social integration indices between early- (<3 years old) and later-orphaned chimpanzees, which further attests to the plausibility of our conjecture. The one measure of indirect connectedness that was slightly higher in the later-orphaned chimpanzees (“reach” in the grooming networks) may reflect that whereas early-orphaned chimpanzees do not seem to be hampered in their engagement to instigate and tolerate social closeness (i.e., proximity), their active involvement in grooming may be somewhat lagging behind. On the other hand, the other four network indices showed no difference at all, so more research would be needed to validate such a conclusion.

According to Vicino and Miller (2015) [53], animals in captivity need ‘Five Opportunities in order to Thrive’ to allow for a good welfare. These are (1) the opportunity for a thoughtfully presented, well-balanced diet; (2) the opportunity to self-maintain; (3) the opportunity for optimal health; (4) the opportunity to express species-specific behaviour and (5) the opportunity for choice and control. At Chimfunshi, the chimpanzees have access to all five of these opportunities. Relatedly, the chimpanzees at Chimfunshi live in spacious environments, which allows the chimpanzees to choose which social partners they want to interact with [54]. In conjunction, these features might have facilitated chimpanzees’ social recovery from their experienced trauma (also see [55]).

While we do not know when exactly the orphans suffered the loss of their mothers, which for some may have been after the sensitive attachment phase, we speculate that the nurturing environment as provided by Chimfunshi (e.g., large, forested enclosures with chimpanzees being able to live in family units; see previous paragraphs) facilitates chimpanzees’ social rehabilitation. These findings and interpretations are consistent with a study in a Japanese sanctuary (the Kumamoto Sanctuary) showing that seven chimpanzees with a traumatic early background did not differ in their grooming behaviour from their captive-born counterparts [16]. The respective authors attributed the chimpanzees’ social rehabilitation to the positive effects of living in a social group. Similar indications can be found in the observations that therapeutic resocialization of chimpanzees who had lost their mother in either early (between 1 and 2 years) or late (between 3 and 4 years) infancy resulted in reduced fecal cortisol metabolite levels ([25] also see [19]), and that orphaned chimpanzees in the wild were found to be no more stressed after two years than chimpanzees whose mother did not die [30]. Furthermore, a comprehensive review indicated that historical improvements in nursery-rearing practices, such as housing young chimpanzees with their peers instead of keeping them isolated and giving them access to outdoors and interactions with adults, may have improved the post-traumatic welfare of chimpanzees, which further attests to the possibly beneficial effects of a nurturing environment on chimpanzees’ potential to overcome detriment [13].

The current study, however, does not suggest that all social trauma suffered by chimpanzees is reversible; rather, our findings indicate that under certain conditions (e.g., trauma happening after sensitive attachment phase, conducive rehabilitation environment) chimpanzees can overcome their social predicament and adjust to typical chimpanzee life. Further studies are needed to understand the precise mechanisms that enable chimpanzees to successfully cope with (different forms of) social trauma.

## 5. Conclusions

The aim of this study was to investigate whether chimpanzees with an early social trauma due to the loss of their mother still suffered social consequences later in their life or if they had recovered from their suffered trauma. In order to determine this, this study looked at orphaned chimpanzees who were habituated into existing groups of chimpanzees living under semi-free ranging conditions at the Chimfunshi Wildlife Orphanage Sanctuary in Zambia.

Based on party size, close proximity and social grooming, we conclude that the orphaned chimpanzees (*n* = 42) were socially indistinguishable from their counterparts who were born and raised in the sanctuary (*n* = 36). Recovery might be due to the nurturing, semi-natural environment and social housing conditions as provided by Chimfunshi. A possible future direction entails studying idiosyncratic behavioural and physiological aspects to investigate social recovery of the orphaned chimpanzees at a higher resolution. Beyond identifying sanctuaries as possibly valuable rehabilitation centres for orphaned chimpanzees, this study demonstrates a remarkable social flexibility in one of our closest living relatives.

## Figures and Tables

**Figure 1 animals-13-00049-f001:**
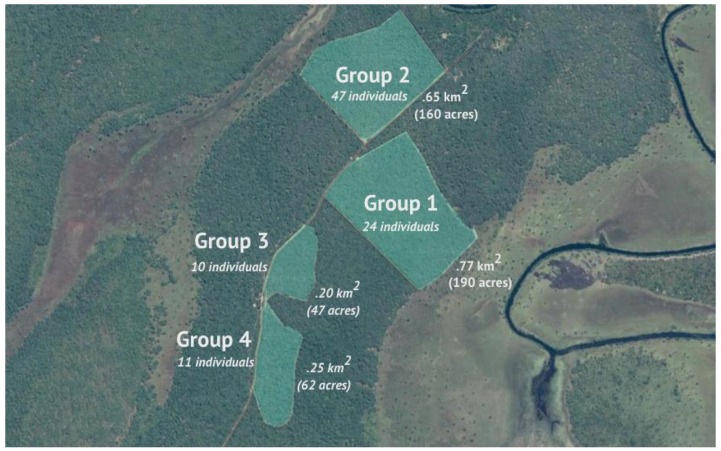
Aerial view of the Project area at the Chimfunshi Wildlife Orphanage Trust in Zambia.

**Figure 2 animals-13-00049-f002:**
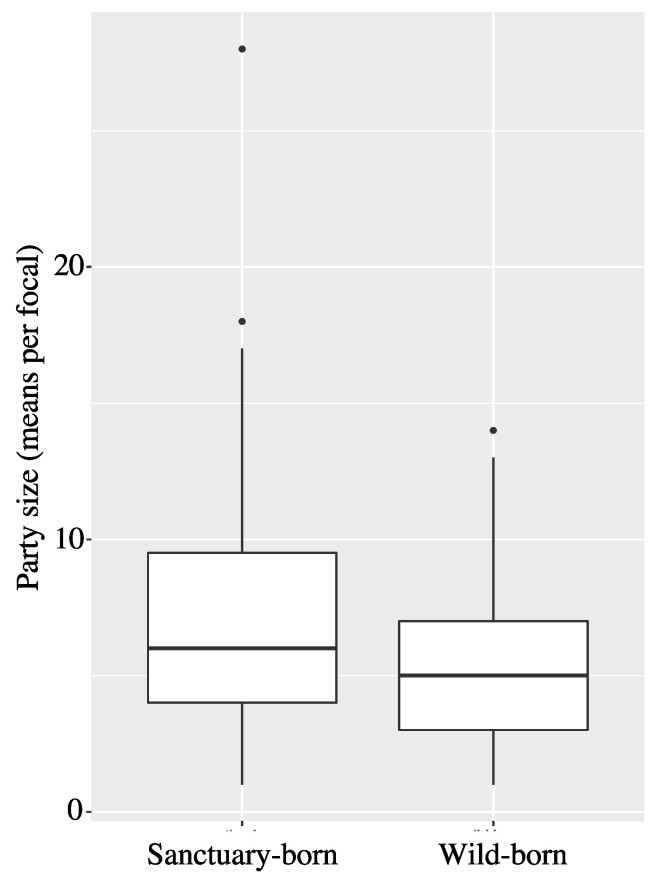
Party sizes for sanctuary-born versus wild-born chimpanzees at the Chimfunshi Wildlife Orphanage Trust, Zambia. Medians are represented by the bold, horizontal lines within the boxes (sanctuary-born = 7.36; wild-born = 6.46). The boxes represent the interquartile range (IQR), the vertical lines attached to the boxes represent Q1 − 1.5 IQR (lower) and Q3 + 1.5 IQR (upper).

**Figure 3 animals-13-00049-f003:**
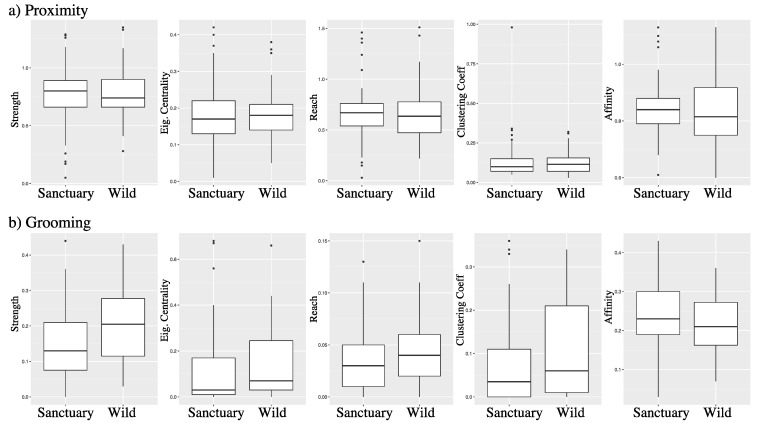
Social network metrics based on (**a**) proximity and (**b**) grooming observations for sanctuary-born versus wild-born chimpanzees. None of the social metrics indicate a significant difference in sociality between chimpanzees with qualitatively dissimilar backgrounds.

**Table 1 animals-13-00049-t001:** Social network metrics (mean ± SD) including significance testing (*p*-value) for the (difference between) sanctuary-born and wild-born chimpanzees at Chimfunshi.

Behavioural Measure	SNA Metric	Sanctuary-Born	Wild-Born	*p*-Value
Proximity	Strength	0.77 ± 0.26	0.78 ± 0.25	0.96
	Eigenvector centrality	0.18 ± 0.09	0.19 ± 0.07	0.99
	Reach	0.67 ± 0.28	0.66 ± 0.27	0.78
	Clustering coefficient	0.14 ± 0.13	0.14 ± 0.08	0.21
	Affinity	0.85 ± 0.10	0.84 ± 0.13	0.11
Grooming	Strength	0.15 ± 0.10	0.21 ± 0.11	0.48
	Eigenvector centrality	0.11 ± 0.16	0.15 ± 0.16	0.07
	Reach	0.04 ± 0.03	0.05 ± 0.03	0.80
	Clustering coefficient	0.07 ± 0.10	0.10 ± 0.10	0.51
	Affinity	0.23 ± 0.09	0.22 ± 0.07	0.09

## Data Availability

The data used in this study are available upon reasonable request.

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
