# Peer review of "Early Trauma Leaves No Social Signature in Sanctuary-Housed Chimpanzees (Pan troglodytes)"

_animals, 2022, doi:10.3390/ani13010049_

Round 1

Reviewer 2 Report

I found this manuscript to be extremely compelling. I also believe that this is timely and important. The study site is well selected and the authors have done a wonderful job explaining aspects of trauma; particularly those due to social deprivation and maternal loss – obviously of keen significance given the population at the study site. To strengthen this introduction even further, I would suggest adding the following reference that would be very relevant:

Bradshaw, G.A., Capaldo, T., Lindner, L. and Grow, G., 2008. Building an inner sanctuary: complex PTSD in chimpanzees. Journal of Trauma & Dissociation, 9(1), pp.9-34.

Chimpanzee spatial needs are really only peripherally mentioned within a description of Chimfunshi (lines 99- 123). Instead, the paper has limited its focus to social dimensions. Lines 113-119 details this focus. While limiting the scope of this to socialization keeps the manuscript sharply focused, I think it leaves out a very important component. Since Chimfunshi provides sanctuary chimpanzees with a large habitat, I believe that looking at the effects of self-determination in mitigating past traumas. 

.Foundational to self-determination is the space of habitat that chimpanzees enjoy at a sanctuary like Chimfunshi. See, for example:

Clark, F.E., 2011. Space to choose: network analysis of social preferences in a captive chimpanzee community, and implications for management. American Journal of Primatology, 73(8), pp.748-757

I think exploring this could be easily accomplished within the existing study design. Including the effects of space on chimpanzee behavior in the introduction would add this as a dimension of mitigating trauma in chimpanzees. While, explaining the use of space that chimpanzees employed during the study would at least give the reader the food for thought to see this as an important aspect of both thriving for chimpanzees and a possible criteria for mitigating past trauma. Without it, one could make the case that chimpanzees with very limited and very managed space, but maternal care or social interactions, could conceivably mitigate past trauma – when in fact, self-determination and free use of space may be highly significant.

A couple of papers that might be worth including in this would be the following:

Duncan, L.M., Jones, M.A., van Lierop, M. and Pillay, N., 2013. Chimpanzees use multiple strategies to limit aggression and stress during spatial density changes. Applied animal behaviour science, 147(1-2), pp.159-171.

Videan, E.N. and Fritz, J., 2007. Effects of short-and long-term changes in spatial density on the social behavior of captive chimpanzees (Pan troglodytes). Applied Animal Behaviour Science, 102(1-2), pp.95-105.

The authors may also do well to familiarize themselves with Vincino and Miller’s “5 opportunities to thrive” for captive animals – as providing opportunities to thrive would be key to mitigating past traumas.

Overall, I believe this is a solid manuscript; and, with the inclusion of spatial needs and self-determination, would be worthy of publication.

Round 2

Reviewer 1 Report

I have no additional comments for the authors, I feel this version of the manuscript is improved with their edits and clarifications.  Thank you. 

Author Response

We are pleased to read that the reviewer feels this version of the manuscript is improved after our editing and clarifications, and that there are no additional comments for us to address.

Thank you for your time, efforts, and consideration.